# Predefined-Performance-Based Full-Process Control for Ultra-Close and High-Precision Formation Flying

Xiande Wu [1], Wenbin Bai [1], Yaen Xie [1,*], Xianliang Zhang [1] and Ting Song [2]

1   College of Aerospace and Civil Engineering, Harbin Engineering University, Harbin 150001, China
2   College of Astronautics, Northwestern Polytechnical University, Xi'an 710072, China
*   Correspondence: xieenya@126.com; Tel.: +86-15776869521

**Abstract:** The prescribed performance robust control method for the leader/follower (L/F) formation is proposed in this paper to solve the problem of spacecraft formation flying (SFF) full-process control (FPC). The objective of FPC is to establish an ultra-close formation with the constraint of collision avoidance between two spacecraft, and then to maintain the formation configuration with high-precision accuracy in a period of time. The main contribution of this paper lies in the following three aspects. Firstly, the six-degree-of-freedom (DOF) error dynamics model of SFF is developed to describe the synchronization motion of the L/F system. Secondly, the prescribed performance bound that comprehensively considers transience and transient performance is designed, which is key for the realization of collision avoidance and high-precision accuracy requirements. Finally, combing prescribed performance control and robust control theories, based on the backstepping method, the predefined performance robust controller is designed, and the tracking errors are proven to converge to the predefined performance bounds in the presence of external disturbances by using the predefined performance robust controller. Illustrative simulations are performed to verify the proposed theoretical results.

**Keywords:** ultra-close formation; predefined performance control; robust control; collision avoidance; formation establishment and maintenance

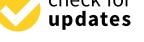



## 1. Introduction

Spacecraft formations can be flown instead of a much bigger and costlier conventional satellite, such as a virtual telescope [1] and distributed antennas [2]. Missions of the spacecraft formation can deliver a comparable or greater mission capability than a monolithic satellite, with significantly enhanced flexibility and robustness [3]. Therefore, the theory of SFF has become the focus of considerably extensive research and development effort during the past two decades [4]. However, in contrast with a monolithic satellite, the main challenge of SFF missions is to maintain precise attitude coordination at a certain relative distance between satellites. Take distributed antennas comprised of L/F formation as an example. In order to achieve high-resolution co-observatories [5], the distance between the edges of the antenna must be maintained within one meter. Meanwhile, the accuracy of the relative position and attitude between satellites should be controlled within the high-precision level. For such SFF missions, collision between satellites needs to be avoided while establishing formation configuration [6]. On the other hand, high-precision accuracy of relative position and attitude needs to be achieved while maintaining formation configuration. The main problem addressed in this paper is that of designing a controller to take an L/F spacecraft formation from some initial configuration to a desired configuration under the condition of avoiding collision, and then to maintain the relative position and attitude between satellites in a high-precision accuracy when the external disturbances are accounted for.

This is undoubtedly a new problem, and SFF full-process control (FPC) of 6-DOF coordination is shown in Figure 1. The FPC of SFF includes two stages of formation control:

formation establishment control and formation maintenance control. Additionally, the 6-DOF coordination control is related primarily to attitude coordination and formation coordination [7]. The previous methods only solved one of the following FPC problems:

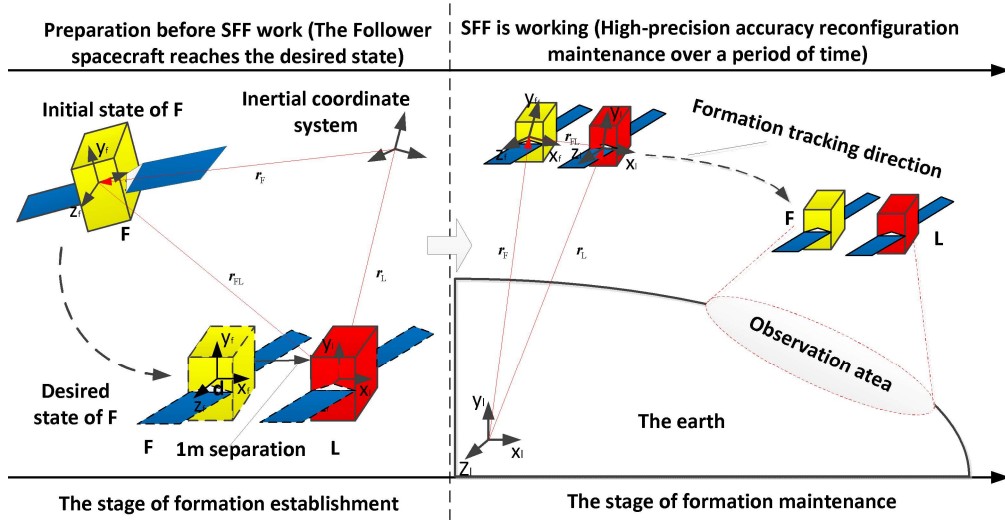

**Figure 1.** Full-process analysis of ultra-close and high-precision SFF tasks.

(1) Formation coordination control and 6-DOF coordination control while establishing formation configuration are summarized here. For example, Ran et al. [7] proposed a robust finite time coordinated formation controller based on the nonsingular terminal sliding mode surface and an adaptive finite time coordinated formation controller by designing a novel sliding mode surface to establish formation configuration. Vaddi et al. [8] developed an analytical two-impulse control scheme to solve spacecraft formation establishment and reconfiguration problems for two-body orbits. Qiu et al. [9] provided a coordinated formation controller considering collision avoidance. In Ref. [10], a distributed control reconfiguration strategy for directed switching topology networked multi-agent systems was developed and investigated in the presence of the loss of effectiveness fault, the outage fault, and the stuck fault. Hua et al. [11] investigated the fault-tolerant time-varying formation control problems for high-order linear multi-agent systems in the presence of actuator failures. Liu et al. [12] proposed an adaptive collision-free formation control strategy for a team of under-actuated spacecraft subject to parametric uncertainties. A distributed 6-DOF coordinated control method of spacecraft formation flying in low earth orbit (LEO) was investigated in Ref. [13]. Sun et al. [14] investigated a 6-DOF spacecraft formation control problem via a composite control method, which consists of a feedback control law based on a finite time control technique and a feedforward compensator based on a nonlinear disturbance observer technique. Huang et al. [15] proposed a collision-free distributed coordination control of 6-DOF spacecraft formation flying. With the estimates provided by the observer, a sliding mode controller was designed for the coordination of 6-DOF formation flying. Meanwhile, the artificial potential function method was employed to design evasive maneuvers in case of any collisions during orbital maneuvering. The above investigations solved the problem of control of spacecraft formation establishment process.

(2) The following paragraph details the attitude coordination control of the SFF. For example, Zhu et al. [16] proposed a robust adaptive coordinated attitude control algorithm for spacecraft formation in the presence of unknown time-varying inertia, persistent external disturbances, and control input saturation. Yang et al. [17] developed a nonlinear attitude tracking control approach for spacecraft formation in the presence of unmeasurable velocity information with time-varying delays and switching topology. In Ref. [18], an attitude coordinated tracking control algorithm for multiple spacecraft formation was investigated with consideration of parametric uncertainties, external disturbances, communication delays,

and actuator saturation. The above investigations solved the problem of 3-DOF attitude control of spacecraft formation.

(3) Formation coordination control and 6-DOF coordination control while maintaining formation configuration is summarized as follows. For example, Agarwal et al. [19] proposed a spacecraft formation maintenance controller that can enable the member spacecrafts to maintain a desired relative orbit with optimal propellant expenditure while maintaining the desired formation. The state feedback controller proposed in [19] was applied on J2 perturbed Clohessy–Wiltshire dynamics, and the system was checked for its stability and performance. For spacecraft formation maintenance, Lee [20] proposed a composed control approach by combining a nonlinear disturbance observer and an asymptotic tracking control for spacecraft formation flying system subject to nonvanishing disturbances. Ran et al. [21] addressed a relative position coordinated control problem for spacecraft formation flying subject to directed communication topology while maintaining formation configuration. Zhao et al. [22] investigated a distributed synchronization control for spacecraft formation with the consideration of unmeasurable modal variable and partial loss of actuator effectiveness faults. Warier et al. [23] proposed a coupled relative attitude and position maintenance control method of a two-spacecraft formation where absolute attitudes are not available. The above investigations solved the problem of control of the spacecraft formation maintenance process.

The above investigations may solve some specific problems for ultra-close and high-precision formation missions, but the goal of this paper is to design a general method for FPC, including formation establishment and formation maintenance, which can achieve collision avoidance between satellites during formation establishment and high-precision accuracy 6-DOF control during formation maintenance.

This goal is accomplished by using a prescribed performance control method [24]. L/F spacecraft formation uses a hierarchical arrangement of individual spacecraft controllers that reduces formation control to individual tracking problems [5]. With the application of predefined performance, the tracking result of the system has been obviously improved since the tracking states of the system are limited in performance bounds. Collision avoidance and tracking accuracy can be achieved by designing performance bounds of relative position and attitude during formation establishment and maintenance. Similarly, predefined-performance control methods have been used by Sui [25] and Yang et al. [26] to improve the tracking accuracy and speed, but these methods only apply to 3-DOF motion and do not consider collision avoidance problems.

Of the previous work, Liu [12] and Lee et al. [27] considered collision-avoidance constraints of the 6-DOF coordination control of SFF using the artificial potential function (APF) method. Calhoun et al. [28] investigated high-precision accuracy control of dual-spacecraft precision formation flying. However, for the ultra-close and high-precision SFF missions, it is necessary to simultaneously consider collision-avoidance constraints during formation establishment and high-precision accuracy control during formation maintenance. In response to the above problems, a predefined performance based FPC method of SFF is proposed in this paper. The main enhancements of this work are summarized as follows:

(1) Full-process of the ultra-close and high-precision SFF is considered, including two stages of formation missions: formation establishment and formation maintenance.

(2) The collision-avoidance problem is explicitly taken into account during formation establishment. By designing performance bounds of relative position between satellites, collision avoidance can be guaranteed.

(3) High-precision accuracy control is taken into account during formation maintenance. Predefined performance control and robust control are combined to ensure 6-DOF coordination in the presence of external disturbances.

The paper is laid out as follows. In Section 2, the kinematics and dynamics models of 6-DOF SFF are developed. Then, a robust backstepping controller with prescribed performance for the ultra-close formation is proposed in Section 3. Finally, the performance

of this control scheme for an ultra-close and high-precision spacecraft formation is evaluated through numerical simulation in Section 4, and the conclusion is given in Section 5.

## 2. Dynamics Model of SFF

Suppose that there is an L/F formation as shown in Figure. 1. The leader is assumed to performs ideal control; that is, the control force and external disturbance force of the leader cancel each other. In this paper, leaders can be regarded as a virtual point. The motion of the virtual point is an ideal orbital motion that is not affected by external forces. The follower is supposed to track the leader to establish formation configuration and then maintain the formation configuration over a period of time. In this section, the motion models of SFF are established.

### 2.1. Coordinate Frames

Three coordinate frames are introduced to establish the motion models of SFF. Their definitions are given as follows.

**Earth-centered inertial (ECI) frame** $F_I = \{O_I x_I y_I z_I\}$. This frame is attached to the earth, where axis $O_I x_I$ points to the vernal equinox, axis $O_I z_I$ points to the north pole, and axis $O_I y_I$ is in the equatorial plane and complies with the right-hand rule.

**Body-centered (BC) frame** $F_i = \{O_i x_i y_i z_i\}$, $i \in \{f, l\}$. This frame, shown in Figure 1, is attached to each spacecraft, where origin $O_i$ is the spacecraft center, and three axes $O_i x_i$, $O_i y_i$ and $O_i z_i$ are along with the inertial principal axes of the spacecraft, respectively.

Local vertical local horizontal (LVLH) frame $F_L = \{O_L x_L y_L z_L\}$. This frame is defined with $O_L y_L$ in the radial direction and pointing in the direction of orbital velocity, $O_L x_L$ in the orbit normal direction and pointing in the opposite direction of the Earth, and $O_L z_L$ completing the right-hand system.

### 2.2. Relative Orbit Motions of SFF

The dynamics models of leader and follower can be expressed as:

$$m_l \ddot{\boldsymbol{R}}_l + m_l \mu \frac{\boldsymbol{R}_l}{||\boldsymbol{R}_l||^3} + \boldsymbol{d}_l = \boldsymbol{F}_l \tag{1}$$

$$m_f (\ddot{\boldsymbol{R}}_l + \ddot{\boldsymbol{R}}_r) + m_f \mu \frac{\boldsymbol{R}_l + \boldsymbol{R}_r}{||\boldsymbol{R}_l + \boldsymbol{R}_r||^3} + \boldsymbol{d}_{ff} = \boldsymbol{F}_f \tag{2}$$

where $m_l$, $m_f$ denote the mass of the leader and follower spacecraft, respectively. $\boldsymbol{R}_l$ is the orbital radius vector of the leader. $\boldsymbol{R}_r$ is the position of the follower spacecraft relative to the leader spacecraft in the ECI frame. $\boldsymbol{d}_l$ and $\boldsymbol{d}_{ff}$ denote the external disturbance acting on the leader and follower, respectively. $\boldsymbol{F}_l$ and $\boldsymbol{F}_f$ is the total force. $\mu = 398600.4418 \left[\text{km}^3/\text{s}^2\right]$ is the gravitational constant.

According to Equations (1) and (2), the relative orbit motion equation can be expressed as follows:

$$m_f \ddot{\boldsymbol{R}}_r + m_f \mu \left( \frac{\boldsymbol{R}_l + \boldsymbol{R}_r}{||\boldsymbol{R}_l + \boldsymbol{R}_r||^3} - \frac{\boldsymbol{R}_l}{||\boldsymbol{R}_l||^3} \right) + \boldsymbol{d}_t = \boldsymbol{F}_t \tag{3}$$

where $\boldsymbol{d}_t = \boldsymbol{d}_f - m_f/m_l \boldsymbol{d}_l$, $\boldsymbol{F}_t = \boldsymbol{F}_f - m_f/m_l \boldsymbol{F}_l$.

It should be noted that $\boldsymbol{R}_r$ is presented in the ECI frame in Equation (3). Define $\boldsymbol{R}_r = \begin{bmatrix} x & y & z \end{bmatrix}^{\text{T}}$ as the relative position in the LVLH frame. Derived from vector $\boldsymbol{R}_r$, we can obtain:

$$\frac{\mathrm{d}\boldsymbol{R}_r}{\mathrm{d}t} = \frac{\delta \boldsymbol{R}_r}{\delta t}\Big|_L + \boldsymbol{\omega} \times \boldsymbol{R}_r \tag{4}$$

where $\delta \boldsymbol{R}_r / \delta t$ indicates the derivative operation of $\boldsymbol{R}_r$ given in the LVLH frame. $\boldsymbol{\omega}$ is the angular velocity of LVLH frame respected to the ECI frame in terms of components along the LVLH frame.

Assuming that the leader is operating along a circular orbit, its orbital angular velocity $\omega_o$ is a constant value. Then, we can obtain:

$$
\begin{aligned}
\frac{d^2 \mathbf{R}_r}{dt^2} &= \frac{\delta^2 \mathbf{R}_r}{\delta t^2} + 2\boldsymbol{\omega} \times \frac{\delta \mathbf{R}_r}{\delta t} + \boldsymbol{\omega} \times (\boldsymbol{\omega} \times \mathbf{R}_r) \\
&= \begin{bmatrix} \ddot{x} \\ \ddot{y} \\ \ddot{z} \end{bmatrix} + 2 \begin{bmatrix} 0 \\ 0 \\ \omega_o \end{bmatrix} \times \begin{bmatrix} \dot{x} \\ \dot{y} \\ \dot{z} \end{bmatrix} + \begin{bmatrix} 0 \\ 0 \\ \omega_o \end{bmatrix} \times \left( \begin{bmatrix} 0 \\ 0 \\ \omega_o \end{bmatrix} \times \begin{bmatrix} x \\ y \\ z \end{bmatrix} \right) = \begin{bmatrix} \ddot{x} - 2\omega_o \dot{y} - \omega_o^2 x \\ \ddot{y} + \omega_o \dot{x} - \omega_o^2 y \\ \ddot{z} \end{bmatrix}
\end{aligned}
\tag{5}
$$

Substituting Equation (5) into Equation (3), we can obtain:

$$
m_f \ddot{\mathbf{R}}_r + \mathbf{C}_t(\omega_o)\dot{\mathbf{R}}_r + N_t(\mathbf{R}_r, \boldsymbol{\omega}, \mathbf{R}_l) + \mathbf{d}_t = \mathbf{F}_t
\tag{6}
$$

where matrix $\mathbf{C}_t(\omega_o)$ and vector $N_t(\mathbf{R}_r, \boldsymbol{\omega}, \mathbf{R}_l)$ are defined as:

$$
\mathbf{C}_t(\omega_o) = 2m_f \omega_o \begin{bmatrix} 0 & -1 & 0 \\ 1 & 0 & 0 \\ 0 & 0 & 0 \end{bmatrix}
\tag{7}
$$

$$
N_t(\mathbf{R}_r, \boldsymbol{\omega}, \mathbf{R}_l) = m_f \begin{bmatrix} \frac{\mu x}{\| \mathbf{R}_l + \mathbf{R}_r \|^3} - \omega_o^2 x \\ \mu \left( \frac{r_l + y}{\| \mathbf{R}_l + \mathbf{R}_r \|^3} - \frac{r_l}{\| \mathbf{R}_l \|^2} \right) - \omega_o^2 y \\ \frac{\mu z}{\| \mathbf{R}_l + \mathbf{R}_r \|^3} \end{bmatrix}
\tag{8}
$$

In leader/follower SFF missions, the leader is assumed to performs ideal control, that is, the control force and external disturbance force of the leader cancel each other, $\mathbf{F}_l = \mathbf{d}_l$. In this paper, leaders can be regarded as a virtual point. The motion of the virtual point is an ideal orbital motion that is not affected by external forces. Then Equation (6) can be equivalently expressed as:

$$
m_f \ddot{\mathbf{R}}_r + \mathbf{C}_t(\omega_o)\dot{\mathbf{R}}_r + N_t(\mathbf{R}_r, \boldsymbol{\omega}, \mathbf{R}_l) + \mathbf{d}_{ff} = \mathbf{F}_f
\tag{9}
$$

*2.3. Relative Attitude Motions of SFF*

**Define** $\mathbf{q}_r = \mathbf{q}_l^{-1} \otimes \mathbf{q}_f$, $\boldsymbol{\omega}_r = \boldsymbol{\omega}_f - \mathbf{C}_r \boldsymbol{\omega}_l$ as the relative attitude quaternion and relative angular velocity of the system, respectively. $\mathbf{q}_l = \begin{bmatrix} \eta_l & \boldsymbol{\varepsilon}_l^{\mathrm{T}} \end{bmatrix}^{\mathrm{T}}$ and $\mathbf{q}_f = \begin{bmatrix} \eta_f & \boldsymbol{\varepsilon}_f^{\mathrm{T}} \end{bmatrix}^{\mathrm{T}}$ are the attitude quaternion of the leader and the follower. $\boldsymbol{\omega}_l$ and $\boldsymbol{\omega}_f$ denote the attitude angular velocity of the leader and the follower represented in the body frame. $\mathbf{C}_r$ is the coordination transformation matrix. The relative attitude kinematic motion of the spacecraft can be expressed as:

$$
\dot{\mathbf{q}}_r = \frac{1}{2} \Xi(\mathbf{q}_r)\boldsymbol{\omega}_r
\tag{10}
$$

where $\Xi(\mathbf{q}_r) = \begin{bmatrix} -\boldsymbol{\varepsilon}_r^{\mathrm{T}} & \eta_r \mathbf{I}_{3\times3} + \mathbf{S}(\boldsymbol{\varepsilon}_r) \end{bmatrix}^{\mathrm{T}}$. $\mathbf{I}_{3\times3}$ denotes the third-order unit matrix.

The relative attitude dynamic motion of the spacecraft is

$$
\mathbf{J}_f \dot{\boldsymbol{\omega}}_r + \boldsymbol{\Lambda}_r \boldsymbol{\omega}_r + N_r + \mathbf{d}_{uf} = \mathbf{u}_f
\tag{11}
$$

The detailed derivation process of Equation (11) is provided in the Appendix A, and

$$
\boldsymbol{\Lambda}_r = \mathbf{S}(\mathbf{C}_r \boldsymbol{\omega}_l)\mathbf{J}_f + \mathbf{J}_f \mathbf{S}(\mathbf{C}_r \boldsymbol{\omega}_l) - \mathbf{S}(\mathbf{J}_f \boldsymbol{\omega}_f)
\tag{12}
$$

$$
N_r = \mathbf{S}(\mathbf{C}_r \boldsymbol{\omega}_l)\mathbf{J}_f \mathbf{C}_r \boldsymbol{\omega}_l - \mathbf{J}_f \mathbf{C}_r \mathbf{J}_l^{-1} \mathbf{S}(\boldsymbol{\omega}_l)\mathbf{J}_l \boldsymbol{\omega}_l
\tag{13}
$$

where $\mathbf{J}_l$ and $\mathbf{J}_f$ denote the inertial matrix of the leader and the follower, respectively. $\mathbf{d}_{uf}$ **denotes the external disturbances acting on the system. Especially for the spacecraft in Earth orbit, $\mathbf{d}_{uf} = \mathbf{d}_{ufg} + \mathbf{d}_{ufo}$. $\mathbf{d}_{ufg}$ and $\mathbf{d}_{ufo}$ denote the gravity gradient moment and other external disturbances, respectively. $\mathbf{u}_f$ denotes the control torque acting on**

**the system.** $C_r$ is the rotation matrix that brings the BC frame $F_l$ onto the BC frame $F_f$, and it can be given as $C_r = (\eta_r^2 - \varepsilon_r^{\mathrm{T}}\varepsilon_r)I_{3\times3} + 2\varepsilon_r\varepsilon_r^{\mathrm{T}} - 2\eta_r S(\varepsilon_r)$. $S(\cdot)$ denotes a $3 \times 3$ skew-symmetric matrix, which is

$$S(\theta) = \begin{bmatrix} 0 & -\theta_3 & \theta_2 \\ \theta_3 & 0 & -\theta_1 \\ -\theta_2 & \theta_1 & 0 \end{bmatrix} \tag{14}$$

*2.4. 6-DOF Motions of SFF*

Define $x_1 = [R_r^{\mathrm{T}} \quad \varepsilon_r^{\mathrm{T}}]^{\mathrm{T}}$, $x_2 = [\dot{R}_r^{\mathrm{T}} \quad \omega_r^{\mathrm{T}}]^{\mathrm{T}}$, according to the Equations (9)–(11), the 6-DOF motions of the SFF can be expressed as:

$$\begin{cases} \dot{x}_1 = \Lambda(x_1)x_2 \\ M_f\dot{x}_2 + C(x_2) + N(x_1, x_2) = u + d \end{cases} \tag{15}$$

where the vectors and matrix can be specifically expressed as:

$$\Lambda = \begin{bmatrix} I_{3\times3} & 0_{3\times3} \\ 0_{3\times3} & 0.5\Xi \end{bmatrix}, \ M_f = \begin{bmatrix} m_f I_{3\times3} & 0_{3\times3} \\ 0_{3\times3} & J_f \end{bmatrix}, \ C = \begin{bmatrix} C_t(\omega_o)\dot{R}_r \\ \Lambda_r\omega_r \end{bmatrix}$$

$$N(x_1, x_2) = \begin{bmatrix} N_t \\ N_r \end{bmatrix}, \ d = \begin{bmatrix} -d_{ff} \\ -d_{uf} \end{bmatrix}, \ u = \begin{bmatrix} F_f \\ u_f \end{bmatrix}$$

*2.5. Problem Statement*

Having derived the kinematics and dynamics models of the relative 6-DOF motion for the system, the control objective is now summarized as eliminating the initial errors with respect to the desired formation configuration in the condition of collision avoidance and thereafter maintaining the formation configuration in a high-precision accuracy for a period of time in the presence of disturbances.

## 3. Full-Process Controller Design
*3.1. Predefined Performance Function*

The following function is selected as the predefined performance function:

$$\rho_i(t) = (\rho_{i0} - \rho_{i\infty})\mathrm{e}^{-l_i t} + \rho_{i\infty}, \ i = 1, \cdots, 8 \tag{16}$$

where $\rho_{i0}$, $\rho_{i\infty}$ and $l_i$ are preset normal numbers. $\rho_{i0}$ is the initial value $\rho_i(t)$, which should be greater than the initial value of the state quantity when selected. $\rho_{i\infty}$ is the final value of the predefined performance function, indicating the maximum allowable steady-state error of the system.

Based on the 6-DOF coordination motion models of SFF in Section 2, the tracking errors of the system can be defined as

$$\begin{cases} e_1 = x_1 - x_d \\ e_2 = x_2 \end{cases} \tag{17}$$

where $e_1 = [e_{11} \quad \cdots \quad e_{16}]^{\mathrm{T}}$. $e_2 = [e_{21} \quad \cdots \quad e_{26}]^{\mathrm{T}}$. $x_d$ is the desired value of $x_1$. As it is stated in [29], such prescribed transient and steady-state tracking error bounds can be satisfied by guaranteeing

$$\underline{\beta}_i\rho_i(t) < e_{1i}(t) < \overline{\beta}_i\rho_i(t), \ i = 1, \cdots, 6 \tag{18}$$

where $\underline{\beta}_i \in [-1, 0]$ and $\overline{\beta}_i \in [0, 1]$ are parameters to be designed. If $\underline{\beta}_i$, $\overline{\beta}_i$, $\rho_{i0}$, $\rho_{i\infty}$ and $l_i$ are selected appropriately, the tracking error is enforced in the allowable region.

In order to achieve collision avoidance, the performance bounds of tracking errors $e_1(t)$, $i = 1, 2, 3$ should be designed to guarantee that there is no overshoot in the stage of formation establishment, as shown in Figure 2. In the stage of formation maintenance, the region of tracking error $e_{1i}(t)$, $i = 1, \cdots, 6$ needs to be within the control accuracy of SFF.

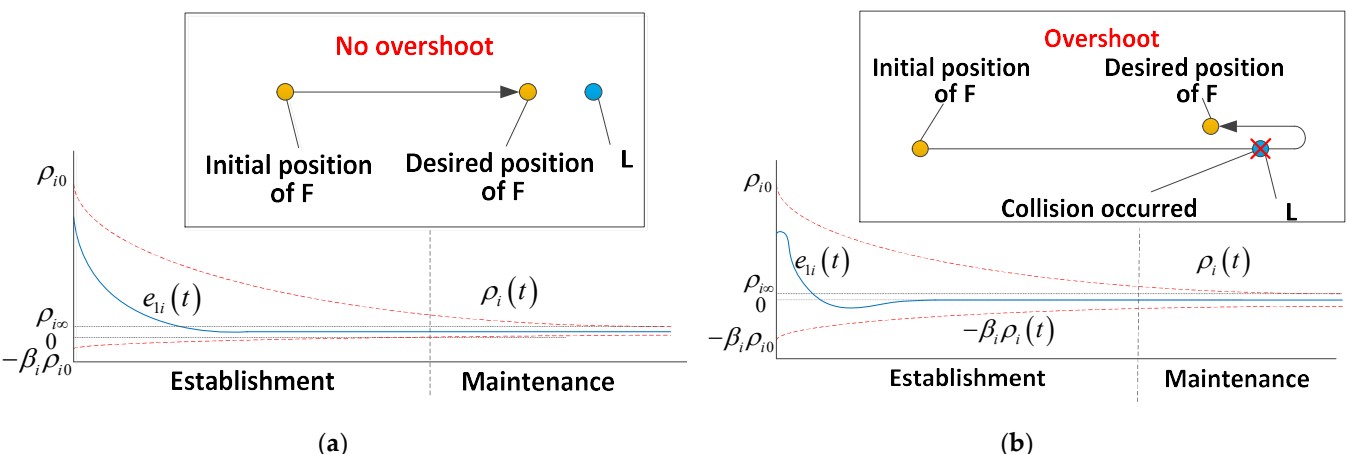

**Figure 2.** Schematic representation of performance bounds. (**a**) Performance bounds design without overshoot. (**b**) Performance bounds design with overshoot.

*3.2. Error Transformation*

The following error transformation is employed to turn the inequality (18) into an equality form

$$e_{1i}(t) = \rho_i(t) N_i(z_i), \ i = 1, \cdots, 6 \tag{19}$$

where $N(\cdot)$ is a smooth, strictly increasing function, and $N(z_i)$ meets the following conditions:

$$\underline{\beta_i} < N_i(z_i) < \overline{\beta_i} \tag{20}$$

Equation (19) can be equivalently transformed as:

$$z_i(t) = N_i^{-1}\left(\frac{e_{1i}(t)}{\rho_i(t)}\right) = T\left(\frac{e_{1i}(t)}{\rho_i(t)}\right), \ i = 1, \cdots, 6 \tag{21}$$

$N_i(z_i)$ is selected with the form as follows:

$$N_i(z_i) = \frac{e^{z_i}\overline{\beta_i} + \underline{\beta_i}}{e^{z_i} + 1}, \ i = 1, \cdots, 6 \tag{22}$$

Then, we have

$$z_i(t) = \ln\left(\frac{e_{1i}}{\rho_i} - \underline{\beta_i}\right) - \ln\left(\overline{\beta_i} - \frac{e_{1i}}{\rho_i}\right) \tag{23}$$

and

$$\dot{z}_i(t) = \frac{\partial N_i^{-1}}{\partial\left(\frac{e_{1i}(t)}{\rho_i(t)}\right)}\left(\frac{e_{1i}(t)}{\rho_i(t)}\right)', i = 1, \ldots, 6 \tag{24}$$

According to Equation (24), the equivalent error models of the 6-DOF coordination SFF can be established as:

$$\begin{cases} \dot{z} = -rv + r\Lambda(x_1)x_2 \\ \dot{x}_2 = f(x_1, x_2) + gu + d \\ y = z \end{cases} \tag{25}$$

where $z = \begin{bmatrix} z_1 & \cdots & z_6 \end{bmatrix}^T$. $g = M_f^{-1}$. $d$ is the external disturbance.

$$r = \text{diag}\left( \frac{\partial N_1^{-1}}{\partial \left( \frac{e_{11}(t)}{\rho_i(t)} \right)} \frac{1}{\rho_i}, \frac{\partial N_2^{-1}}{\partial \left( \frac{e_{12}(t)}{\rho_i(t)} \right)} \frac{1}{\rho_i}, \cdots, \frac{\partial N_6^{-1}}{\partial \left( \frac{e_{16}(t)}{\rho_i(t)} \right)} \frac{1}{\rho_i} \right) \tag{26}$$

$$v = \left[ \frac{e_{1i}(t)}{\rho_i(t)}, \frac{e_{2i}(t)}{\rho_i(t)}, \cdots, \frac{e_{6i}(t)}{\rho_i(t)} \right]^T \tag{27}$$

$$f(x_1, x_2) = M_f^{-1}(-C(x_2) - N(x_1, x_2)) \tag{28}$$

*3.3. Controller Design*

In order to achieve high-precision accuracy of relative position and relative attitude under the condition of external disturbances, a predefined performance robust controller is designed in this section, which combines the predefined performance control method with robust control method.

**Definition:** *For the system (25), if there are control law **u** and Lyapunov function V, and the L2 gain from the disturbance d to the control output **y** is less than a given normal number $\gamma$, which is* [30]

$$V(t) - V(0) \le \int_0^T (\gamma^2 ||d||^2 - ||y||^2) dt \tag{29}$$

*where $T > 0$, then the system is said to be internally stable with robust $H_\infty$ disturbance suppression performance.*

In what follows, we begin the predefined performance robust controller design for system (25) based on backstepping.

Select the virtual variable as follows:

$$\begin{cases} \tilde{\xi}_1 = z \\ \tilde{\xi}_2 = e_2 - \alpha_1 \end{cases} \tag{30}$$

We choose the following Lyapunov function candidate:

$$V_1 = \frac{1}{2} \tilde{\xi}_1^T \tilde{\xi}_1 \tag{31}$$

The time derivative of $V_1$ is obtained as:

$$\begin{aligned} \dot{V}_1 &= \tilde{\xi}_1^T \dot{\tilde{\xi}}_1 \\ &= \tilde{\xi}_1^T [-rv + r\Lambda(x_1)x_2] \end{aligned} \tag{32}$$

In order to stabilize $\tilde{\xi}_1$, we choose $\alpha_1$ as:

$$\alpha_1 = -\frac{1}{r\Lambda(x_1)} K_1 \tilde{\xi}_1 + \frac{v}{\Lambda(x_1)} \tag{33}$$

where $K_1$ is a positive definite matrix.

Substituting Equation (33) into Equation (32), we can obtain:

$$\dot{V}_1 = -\tilde{\xi}_1^T K_1 \tilde{\xi}_1 \tag{34}$$

whereafter, consider the following Lyapunov function candidate:

$$V_2 = \frac{1}{2} \tilde{\xi}_1^T \tilde{\xi}_1 + \frac{1}{2} \tilde{\xi}_2^T \tilde{\xi}_2 \tag{35}$$

and

$$H = \dot{V}_2 + \frac{1}{2}||\boldsymbol{y}||^2 - \frac{\gamma^2}{2}||\boldsymbol{d}||^2 \tag{36}$$

where $\gamma$ is a positive constant.

Substituting Equation (35) into Equation (36), we can obtain:

$$
\begin{aligned}
H &= \xi_1^{\mathrm{T}}\dot{\xi}_1 + \xi_2^{\mathrm{T}}\dot{\xi}_2 + \frac{1}{2}||\boldsymbol{y}||^2 - \frac{\gamma^2}{2}||\boldsymbol{d}||^2 \\
&= -\xi_1^{\mathrm{T}}\boldsymbol{K}_1\xi_1 + z_2^{\mathrm{T}}(\boldsymbol{f} + \boldsymbol{\Lambda}\boldsymbol{u} + \boldsymbol{d} - \dot{\boldsymbol{\alpha}}_1) + \frac{1}{2}||\boldsymbol{y}||^2 - \frac{\gamma^2}{2}||\boldsymbol{d}||^2
\end{aligned}
\tag{37}
$$

In order to stabilize $z_1$ and $z_2$, the controller can be designed as:

$$\boldsymbol{u} = \boldsymbol{g}^{-1}(-\frac{z_2}{\gamma^2} - \boldsymbol{f} - \boldsymbol{K}_2\xi_2 + \dot{\boldsymbol{\alpha}}_1) \tag{38}$$

Substituting Equation (38) into Equation (37), we can obtain:

$$
\begin{aligned}
H &= -\xi_1^{\mathrm{T}}\boldsymbol{K}_1\xi_1 + \xi_2^{\mathrm{T}}(-\frac{\xi_2}{\gamma^2} - \boldsymbol{K}_2\xi_2 + \boldsymbol{d}) + \frac{1}{2}||\boldsymbol{y}||^2 - \frac{\gamma^2}{2}||\boldsymbol{d}||^2 \\
&= -\xi_1^{\mathrm{T}}\boldsymbol{K}_1\xi_1 - \xi_2^{\mathrm{T}}\boldsymbol{K}_2\xi_2 - \frac{\xi_2^{\mathrm{T}}\xi_2}{\gamma^2} + \xi_2^{\mathrm{T}}\boldsymbol{d} + \frac{1}{2}||\boldsymbol{y}||^2 - \frac{\gamma^2}{2}||\boldsymbol{d}||^2 \\
&= -\xi_1^{\mathrm{T}}(\boldsymbol{K}_1 - \frac{1}{2}\boldsymbol{I})\xi_1 - \xi_2^{\mathrm{T}}\boldsymbol{K}_2\xi_2 - \left(\frac{\gamma}{2}\boldsymbol{d} - \frac{\xi_2}{\gamma}\right)^2 - \frac{\gamma^2}{4}||\boldsymbol{d}||^2
\end{aligned}
\tag{39}
$$

Choose the appropriate $\boldsymbol{K}_1$, $\boldsymbol{K}_2$ to guarantee that $(\boldsymbol{K}_1 - \frac{1}{2}\boldsymbol{I})$ and $\boldsymbol{K}_2$ are semi-definite matrix, then $H < 0$ can be obtained.

Subsequently, the following Lyapunov function candidate is constructed:

$$V = 2V_2 \tag{40}$$

The derivative of $V$ can be expressed as:

$$\dot{V} = 2H - \gamma^2(||\boldsymbol{y}||^2 - ||\boldsymbol{d}||^2) \tag{41}$$

Since $H < 0$, we can obtain:

$$\dot{V} < \gamma^2(||\boldsymbol{d}||^2 - ||\boldsymbol{y}||^2) \tag{42}$$

Equation (42) shows that the robust control algorithm derived from the Lyapunov function based on the backstepping technique makes the error system stable for all bounded disturbances. In summary, the control law can be used to ensure that the closed-loop system error is bounded, thus meeting the predefined transient and steady-state requirements.

## 4. Simulation Results

In this section, we present numerical simulations to illustrate the effectiveness of the proposed FPC. The L/F formation is assumed to consist of two identical antenna spacecraft, as shown in Figure 1. The size of the antenna array of each spacecraft is 2 m × 0.8 m. The parameters of this ultra-close formation flying system and the initial conditions are listed in Table 1. The desired state is given by $x_d = \begin{bmatrix} 1 & 0 & 0 & 0 & 0 & 0 \end{bmatrix}^T$ and the external disturbance input is

$$
\boldsymbol{d} = \begin{bmatrix} -\boldsymbol{d}_{ff} \\ -\boldsymbol{d}_{uf} \end{bmatrix}, \ \boldsymbol{d}_{ff} = -\begin{bmatrix} 0.0001 - 0.0001\sin(\pi t/200) \\ 0.0001 - 0.0001\cos(\pi t/200) \\ 0.0001 + 0.0001\sin(\pi t/200) \end{bmatrix} \mathrm{N}, \ \boldsymbol{d}_{uf} = -\begin{bmatrix} 0.001 - 0.001\sin(\pi t/200) \\ 0.001 - 0.001\cos(\pi t/200) \\ 0.001 + 0.001\sin(\pi t/200) \end{bmatrix} \mathrm{N}\cdot\mathrm{m}
$$

which is applied to the system simulation. The total simulation time is about 800 s.

**Table 1.** System parameters for simulation.

| Parameters of Mass and Inertial Matrix | Orbital Parameters | Initial States |
|---|---|---|
| $m_l = 109 \ \text{kg} \ m_f = 109 \ \text{kg}$ $J_f = \text{diag}\{18.9, 18.9, 26.2\}$ $J_l = \text{diag}\{18.9, 18.9, 26.2\}$ | $R_l = 7578.17\text{km}$ $\omega_0 = 9.5702 \times 10^{-4} \ \text{rad/s}$ | $\boldsymbol{R}_r(0) = \begin{bmatrix} 20 & 28 & 15 \end{bmatrix}^{\text{T}} \text{m}$ $\boldsymbol{v}_r(0) = \begin{bmatrix} 0 & 0 & 0 \end{bmatrix}^{\text{T}} \text{m/s}$ $\boldsymbol{\omega}_r(0) = \begin{bmatrix} 0.001 & 0.002 & -0.001 \end{bmatrix}^{\text{T}} \ \text{rad/s}$ $\boldsymbol{q}_r(0) = \begin{bmatrix} 0.707 & 0.4 & 0.5 & 0.3 \end{bmatrix}^{\text{T}}$ |

The predefined performance robust controller is designed through the procedures given in Section 3. According to the FPC requirement of high precision and rapid convergence, the prescribed performances of the relative attitude quaternion are set to steady error of no more than $\rho_{i\infty} = 10^{-3}(i = 1, 2, 3)$, minimum convergence speed $l_i = 0.01(i = 1, 2, 3)$, and the parameters related to overshoot $\overline{\beta}_i = 1(i = 1, 2, 3)$. $\underline{\beta}_i = 0(i = 1, 2, 3)$. Similarly, the prescribed performances of the relative position are set to steady error of no more than $\rho_{i\infty} = 10^{-3}(i = 4, 5, 6)$, minimum convergence speed $l_i = 0.016(i = 4, 5, 6)$. Further, considering the anti-collision constraint and initial states, the parameters related to overshoot of the predefined performance control for the relative position are chosen as $\overline{\beta}_4 = 1, \overline{\beta}_5 = 1, \overline{\beta}_6 = 0, \underline{\beta}_4 = 0, \underline{\beta}_5 = 0, \underline{\beta}_6 = -1$. The controller parameters are shown in Table 2. Based on the above parameters, the simulation results are shown as follows.

**Table 2.** Parameters for controllers.

| Controller Gains |
|---|
| $\boldsymbol{K}_1 = 0.5\boldsymbol{I}_{6\times6}$ |
| $\boldsymbol{K}_2 = \text{diag}(0.3, 0.3, 0.3, 0.4, 0.4, 0.4)$ |
| $\gamma = 10 \ \boldsymbol{J}_l = \text{diag}\{18.9, 18.9, 26.2\}$ |

Figure 3 is the relative attitude quaternion between the leader and follower. Figure 4 is the relative angular velocity between the leader and follower. Figure 5 is the control torque acting on the follower. Figure 6 is the relative position between the leader and follower. Figure 7 denotes the 3-DOF relative motion trajectories between the leader and follower. Figure 8 is the relative orbital velocity between the leader and follower. Figure 9 is the control force acting on the follower.

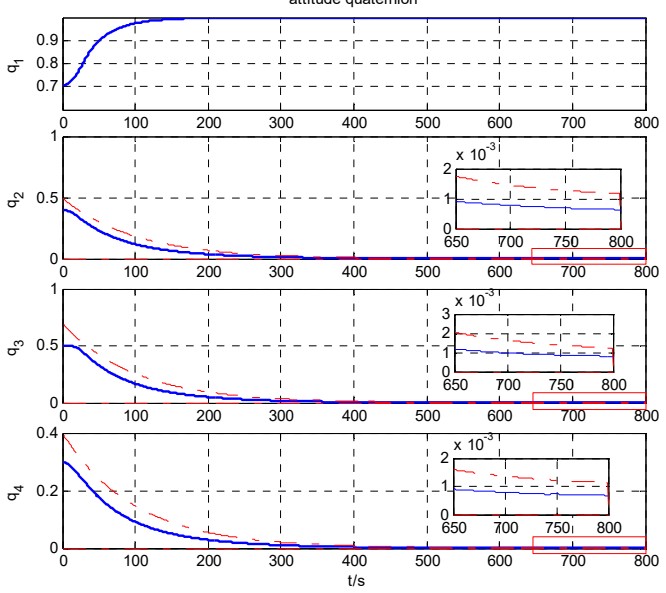

**Figure 3.** Trajectories of relative attitude quaternion.

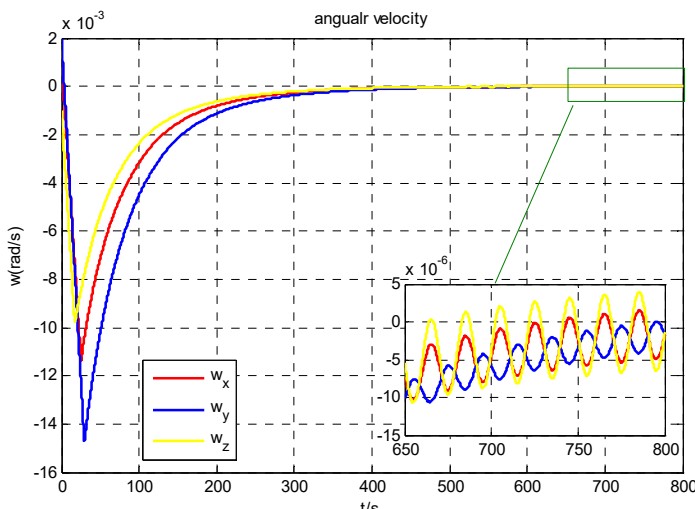

**Figure 4.** Trajectories of relative angular velocity.

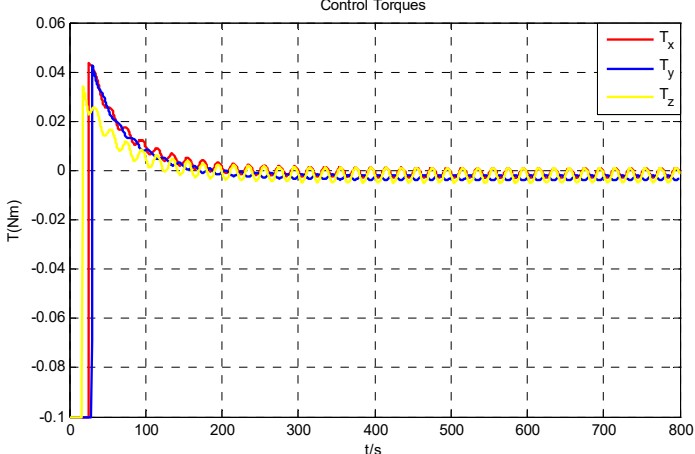

**Figure 5.** Applied torques.

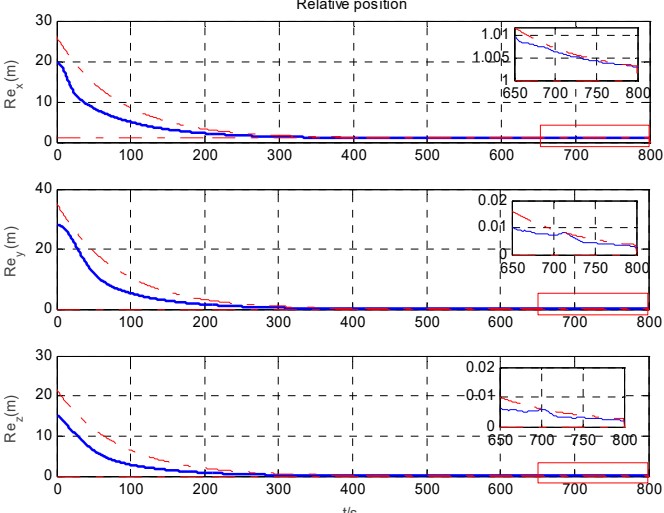

**Figure 6.** Trajectories of relative position.

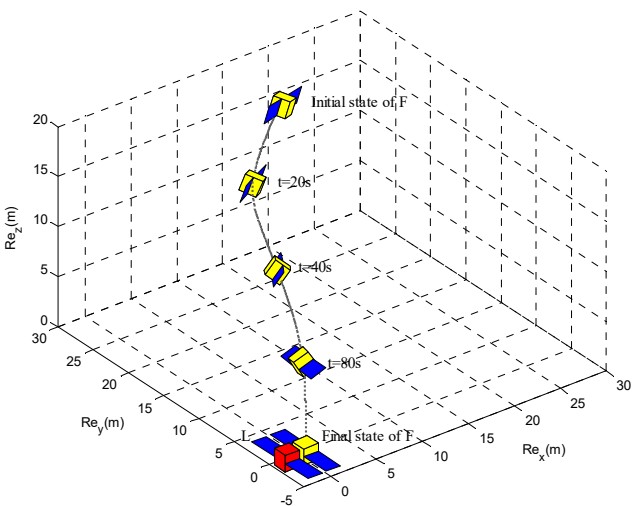

**Figure 7.** The process of the formation establishment.

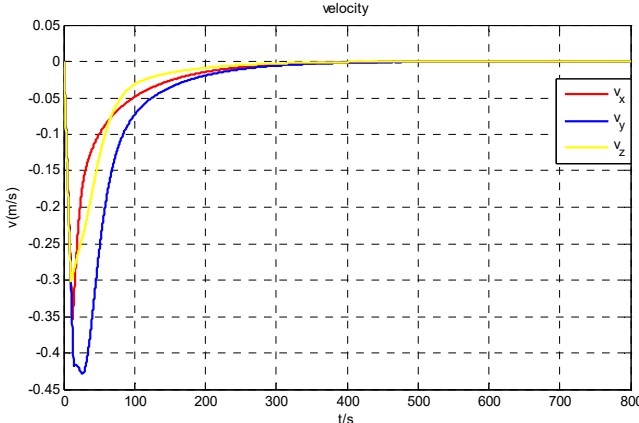

**Figure 8.** Trajectories of relative velocity.

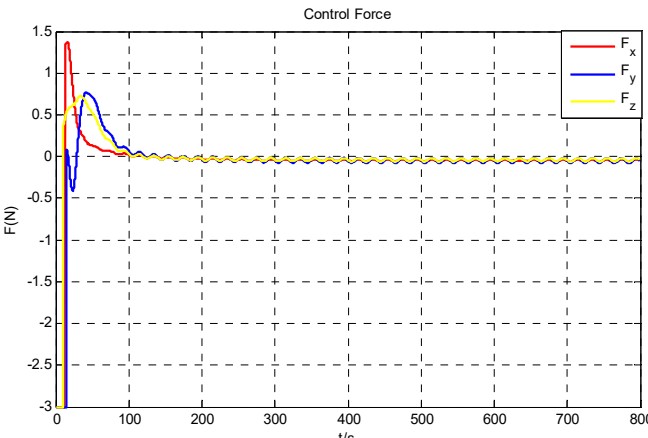

**Figure 9.** Applied forces.

Figure 3 shows that the relative attitude quaternions of the formation, kept inside the desired predefined performance bounds, clearly satisfy the prescribed performance specifications. It can be observed that the attitude synchronization of the L/F formation is realized in high-precision accuracy. Figure 4 shows that all the relative angular velocities converge to zero asymptotically. Figure 5 presents the control torques for the motion of relative attitude of the formation. The initial control efforts are relatively large in order

to drive the system to the desired states quickly. They decrease rapidly after the desired values are achieved. Very small torques are required to maintain the relative attitude of the formation due to the disturbances. The amplitude of torques required by the controller is less than 0.1 Nm, which is reasonable and can be implemented through conventional angular momentum exchange device.

Figure 6 shows that all the relative positions are kept inside the desired predefined performance bounds, and there is no overshoot in the trajectory of $R_r$. Therefore, collision-avoidance between spacecraft is achieved. Figure 7 shows the process of the formation establishment. The arrows indicate the attitude of the x-axis in the body coordinate system of the follower spacecraft. It is found that the attitude and position of the follower spacecraft reach the desired states; that is, the relative attitude of the two spacecraft is consistent, and the relative distance between two spacecrafts is maintained at 1 m, realizing the control requirement under the proposed robust controller. Moreover, as shown in Figures 6 and 8, all the relative positions and velocities converge to the neighborhoods of the desired states asymptotically. It is proven that high-precision control accuracy is implemented under the condition of external disturbances, which shows the robustness of the proposed control strategy. Figure 9 presents the control forces for the motion of relative positions of the formation. The control efforts decrease rapidly after the desired distance between the two spacecraft are achieved. Very small forces are required to maintain the relative positions of the formation due to the disturbances. It can be observed that the control forces are on the order of 3N, which can be implemented by thrusters.

## 5. Conclusions

This paper investigates the FPC problems of ultra-close and high-precision spacecraft formations. A predefined performance robust control scheme was developed. On the one hand, by designing predefined performance transient-state bounds of tracking position errors, collision-avoidance is implemented during formation establishment. On the other hand, high-precision accuracy control was achieved by keeping all tracking errors inside the desired predefined performance steady-state bounds. In addition, robust algorithms were developed to update the performance of disturbance rejection. The boundedness and convergence of the closed-loop system were confirmed based on the Lyapunov stability theory. Finally, numerical simulation was conducted to control the SFF comprised of two identical antenna spacecraft. According to the results of the simulation, the proposed controller can successfully achieve formation establishment and high-precision formation maintenance, owing to external disturbances, while the constraint of the collision-avoidance is satisfied. As future works, experimental implementation of the FPC method will be investigated using semi-physical simulators.

**Author Contributions:** Conceptualization, X.W. and Y.X.; methodology, X.W. and W.B.; software, W.B. and X.Z.; validation, W.B.; formal analysis, T.S.; investigation, T.S.; resources, Y.X.; data curation, W.B.; writing—original draft preparation, W.B.; writing—review and editing, X.W. and Y.X. All authors have read and agreed to the published version of the manuscript.

**Funding:** This research was funded by the Basic Research Project of the Science and Technology on Complex Electronic System Simulation Laboratory, grant number: DXZT-JC-ZZ-2020-012, the National Natural Science Foundation of China, grant number: No. 11772185, the project, grant number: 020214, and the Fundamental Research Foundation of the Central Universities, grant number: 3072022TS0401, 3072022CFJ0202, 3072022CFJ0204.

**Data Availability Statement:** Not applicable.

**Conflicts of Interest:** We confirm that this manuscript has not been published elsewhere and is not under consideration by another journal. All authors have approved the manuscript and agree with its submission to Aerospace.

## Appendix A

The specific derivation process of Equations (11)–(13) is as follows.

The attitude dynamic equation of the leader can be expressed as:

$$J_l\dot{\omega}_l + \omega_l \times J_l\omega_l + d_{ul} = u_l \tag{A1}$$

where $\omega_l$, $u_l$ and $d_{ul}$ denote the angular velocity, control torque and external disturbance torque acting on the leader with respect to the BC frame F$_l$. In leader/follower SFF missions, the leader is always assumed to performs ideal control, that is, the control torque and external disturbance torque of the leader cancel each other, $u_l = d_{ul}$.

The attitude dynamic equation of the follower can be expressed as:

$$J_f\dot{\omega}_f + \omega_f \times J_f\omega_f + d_{uf} = u_f \tag{A2}$$

where $\omega_f$, $u_f$ and $d_{uf}$ denote the angular velocity, control torque, and external disturbance torque acting on the follower with respect to the BC frame F$_f$.

The relative angular velocity between the leader and follower can be expressed as:

$$\omega_r = \omega_f - C_r\omega_l \tag{A3}$$

Solving the first derivative of Equation (45), one can obtain:

$$\dot{\omega}_r = \dot{\omega}_f + \omega_r \times \omega_f - C_r\dot{\omega}_l \tag{A4}$$

The solution of Equation (46) refers to the calculation steps in Ref. [31]. Multiplying the resultant expression by $J_f$ on both sides, we obtain:

$$\begin{aligned}
J_f\dot{\omega}_r &= J_f\dot{\omega}_f + J_f S(\omega_r)\omega_f - J_f C_r\dot{\omega}_l \\
&= J_f\dot{\omega}_f + J_f S(\omega_r)(\omega_r + C_r\omega_l) - J_f C_r\dot{\omega}_l \\
&= J_f\dot{\omega}_f + J_f S(\omega_r)(C_r\omega_l) - J_f C_r\dot{\omega}_l
\end{aligned} \tag{A5}$$

Substituting Equations (43) and (44) into Equation (47), we can obtain:

$$\begin{aligned}
J_f\dot{\omega}_r &= J_f\dot{\omega}_f + J_f S(\omega_r)C_r\omega_l - J_f C_r\dot{\omega}_l \\
&= -S(\omega_r + C_r\omega_l)J_f(\omega_r + C_r\omega_l) + u_f + d_{uf} + J_f S(\omega_r)C_r\omega_l - J_f C_r J_l^{-1}(-S(\omega_l)J_l\omega_l) \\
&= -\left[S(\omega_r)J_f\omega_r + S(\omega_r)J_f(C_r\omega_l) + S(C_r\omega_l)J_f\omega_r + S(C_r\omega_l)J_f(C_r\omega_l)\right] - u_f + d_{uf} \\
&\quad + J_f S(\omega_r)(C_r\omega_l) + J_f C_r J_l^{-1}S(\omega_l)J_l\omega_l \\
&= S\left(J_f\omega_r\right)\omega_r - S(C_r\omega_l)J_f\omega_r + S\left(J_f(C_r\omega_l)\right)\omega_r - J_f S(C_r\omega_l)\omega_r \\
&\quad - S(C_r\omega_l)J_f(C_r\omega_l) + J_f C_r J_l^{-1}S(\omega_l)J_l\omega_l - u_f + d_{uf} \\
&= \left[S\left(J_f\omega_f\right) - S(C_r\omega_l)J_f - J_f S(C_r\omega_l)\right]\omega_r - S(C_r\omega_l)J_f(C_r\omega_l) + J_f C_r J_l^{-1}S(\omega_l)J_l\omega_l \\
&\quad - u_f + d_{uf}
\end{aligned} \tag{A6}$$

Equations (11)–(13) can be proved.

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
