# Peer review of "Predefined-Performance-Based Full-Process Control for Ultra-Close and High-Precision Formation Flying"

_aerospace, doi:10.3390/aerospace10020152_

Round 1

Reviewer 1 Report

The work is undoubtedly interesting, original and worthy of publication. It should be especially noted that the authors managed to solve the problem of both creating and maintaining a given SFF configuration from a unified standpoint, and in this problem the satellites are located at a very close distance from each other. Consideration of such SFF also highlights this work. Authors should be encouraged to take on such tasks. However, a few remarks can be made. The authors claim that the problem of the simultaneous initialization and maintenance of the SFF has been solved for the first time. However, they failed to notice a 520-page book [1] that deals with both of these problems. In addition, this book contains a general theory of maneuvering in near circular orbits, which is useful for specialists who solve problems of this type. This book also contains the derivation of the equations of relative motion, the authors make this conclusion in the first part of their work, although these equations have long been known. In the book, deviations are given in a cylindrical coordinate system, which is much more convenient for understanding the physics of maneuvers and calculating deviations along the orbit. The authors are proud that they have solved the problem in what they think is a general setting, but this is far from being the case. They do not take into account errors in the implementation of control, and in the several papers of Boutonnet A., unnoticed by them, it is shown that in order to avoid a collision, it is necessary not only to compensate for errors in the execution of maneuvers that transfer the spacecraft to a given position relative to the leader, but even to compensate for errors in maneuvers that compensate for errors. The authors consider a problem in which the leader does not maneuver, but this option is extremely rare. If both spacecraft maneuver simultaneously, then control errors will significantly complicate the task of eliminating a possible collision. The control itself proposed by the authors is extremely complex and it is unlikely that it can be implemented on a real spacecraft. For real spacecraft, the simpler the better (more reliable, etc.). I say this as a specialist who himself calculated the maneuvers of about 140 real spacecraft, and whose algorithms were used in France and China for real spacecraft. It is not clear where the constants included in the control come from and why they are dimensionless. It is forgotten that the SFF will have to avoid a possible collision with one or several space debris objects at the same time, then both spacecraft will have to maneuver in order to maintain the given configuration. This problem has already been solved by a group of authors. But it is not clear how the method of calculating the control proposed by the authors will work in this case. I'm not an English specialist, but I don't like the combination of forming a formation. In my articles, I try to avoid it. I think it's better to use the term "deployment", naturally, I do not insist on this remark.

  1. Andrey Baranov Spacecraft Manoeuvring in the Vicinity of a Near-Circular Orbit. Cambridge Scholars Publishing, 2022. 520 p

Author Response

Thank you for your comments concerning our manuscript entitled “Predefined Performance Based Full-process Control for Ultra-close and High-precision Formation Flying”. Those comments are all valuable and very helpful for revising and improving our paper, as well as the important guiding significance to our researches. In addition, we are very interested in your research content and reference materials. Thank you for recommending the book written by Andrey Baranov. We plan to buy one and use it to guide our future research direction. Thanks again for your valuable comments.

Reviewer 2 Report

First of all it is not clear for what kind of reasons "ultra-close" formation flying is needed. Is this small relative distance needed for some specific scientific mission? Or for commercial one? Because while the goal of this FF is not stood there is no point in this study. 

Eq.(5) is supposed to be written in projection on LVLH frame, but projection of angular velocity ω doesn't match the description of LVLH frame. 

In lines 130-131 it is said that "the leader is operating along a circular orbit, its orbital angular velocity is a constant value". Being on a circular orbit means not only constant angular velocity but also the absence of disturbances. If you want to include disturbances then angular acceleration must be added in eq.5. 

Also, there is a problem in quaternion kinematics. If the relative attitude quaternion q_r is written with respect to leader body frame, then the quaternion multiplication in line 137 is written in the wrong order. It must be q_r = (q_l)^(-1) × q_f. 

Eq.(11-13) are incorrect, they can't be derived from equations of motion. Hence, the remaining part of the paper, including control law derivation, must be rewritten. 

In angular motion model gravity gradient torque is not included. It can be considered as a disturbance but should be presented. 

In line 230 desired state is given by x_d = [0 0 0 1 0 0], it means that R relative desired is [0 0 0]. It doesn't match with the task. 

Author Response

Thank you for your comments concerning our manuscript entitled “Predefined Performance Based Full-process Control for Ultra-close and High-precision Formation Flying”. Those comments are all valuable and very helpful for revising and improving our paper, as well as the important guiding significance to our researches. We have studied comments carefully and have made correction which we hope meet with approval. Revised portion are marked in bold font with red color in the paper. The main corrections in the paper and the responds to the reviewers’ comments are as flowing:

Response to the comment: First of all it is not clear for what kind of reasons "ultra-close" formation flying is needed. Is this small relative distance needed for some specific scientific mission? Or for commercial one? Because while the goal of this FF is not stood there is no point in this study.

Response: Thanks for your comments. Your comments are sincere and professional. According to our research, in some distributed space missions, multiple spacecraft are required to form a formation to build a super large structure. The edge distance of the load carried by these spacecrafts can be as close as one meter. For example, Sun proposed a distributed radar mission in "Dual-Quaternion-Based Translation-Rotation-Vibration Integrated Dynamics Modeling for Flexible Spacecraft (doi: 10.1061/(ASCE)AS.1943-5525.0000969)" with the edge distance of radar load as close as one meter.

Response to the comment: Eq.(5) is supposed to be written in projection on LVLH frame, but projection of angular velocity ω doesn't match the description of LVLH frame.

Response: Thanks for your comments. In response to your question, we have deduced Eq. (5) again. The definition of the LVLH frame in this paper is as follows: Local vertical local horizontal (LVLH) frame . This frame is defined with  in the radial direction, in the orbit normal direction, and  completing the right-hand system. The projection of angular velocity ω in LVLH frame is around the z-axis. Please review it again. If there is any error, please specify it. Thank you again for your review.

Response to the comment: In lines 130-131 it is said that "the leader is operating along a circular orbit, its orbital angular velocity is a constant value". Being on a circular orbit means not only constant angular velocity but also the absence of disturbances. If you want to include disturbances then angular acceleration must be added in eq.5.

Response: Thanks for your comments. In the original manuscript, our description here is not clear enough, causing misunderstanding. In fact, spacecraft in any orbit will be subject to external interference. The reason why the spacecraft can maintain in the circular orbit is that the spacecraft is launched into the circular orbit, and the control force of the spacecraft and the external interference force on the circular orbit cancel each other. According to your comments, we have modified the description here in the “revised manuscripts”. The revised portion are marked in bold font with red color.

The above modifications are on page 4 (lines 134-136).

Response to the comment: Also, there is a problem in quaternion kinematics. If the relative attitude quaternion q_r is written with respect to leader body frame, then the quaternion multiplication in line 137 is written in the wrong order. It must be q_r = (q_l)^(-1) × q_f.

Response: Thanks for your comments. According to your suggestion, we have modified the equation here.

The above modifications are on page 4 (line 138).

Response to the comment: Eq.(11-13) are incorrect, they can't be derived from equations of motion. Hence, the remaining part of the paper, including control law derivation, must be rewritten.

Response: We have derived Eqs. (11-13) again. The detailed derivation process is provided in the Appendix. We agreed that there was no derivation error. Please review it again. If there is any derivation error, please point out the specific steps of the derivation error.

The above modifications are on pages 13 and 14 (lines 326-341).

Response to the comment: In angular motion model gravity gradient torque is not included. It can be considered as a disturbance but should be presented.

Response: Thanks for your comments. According to your suggestion, we added the explanation of external interference torque in the “revised manuscripts”. The revised portion are marked in bold font with red color.

The above modifications are on page 5 (lines 148-151).

Response to the comment: In line 230 desired state is given by x_d = [0 0 0 1 0 0], it means that R relative desired is [0 0 0]. It doesn't match with the task.

Response: Thanks for your comments. The desired state is given by x_d = [0 0 0 1 0 0], it means that R relative desired is [1 0 0].

Thanks again for your valuable comments.

Round 2

Reviewer 2 Report

Defenition LVLH frame from your work lines 158-160: "This frame is defined with Oy in the radial direction, Ox in the orbit normal direction, and Oz completing the right-hand system." Then we look to the eq.(5) and see that angular velocity ω that supposed to be written in projection on LVLH frame have components along the Oz axes but angular velocity must be in the orbit normal direction (which is Ox axes in your defenition). This have to be corrected.

In line 282 desired state is given by x_d = [0 0 0 1 0 0]. Then I look to the eq.(17) where x_d appears for the first time and it is written that e1 = x1 - x_d, so, this means that x_d have the same dimension as x1. The definition of x1 is written in line 202: x1 = [R_r, ε_r]. This means that first three components of x_d have to correspond to desired relative R. And it is written that they are [0 0 0], so, how this can be?

Tnank you for providing detailed derivation process for eq.(11)-(13) in the Appendix. It clarified some issues but also new questions have been raised. In the first version of the article leader satellite was considered to be uncontrolled, in the latest version it is standed that leader satellite has ideal control that eliminate all external forces. This is quite strong assumption even for a preliminary study. Can you add some disturbances for leader satellite just in the simulation? Or can you comment how are you going to achieve ideal control that cancels all disturbances?

Thank you for providing an example with a distributed radar mission for the explanation of the goal of ultra close formation flying, but I still have some issues about the problem statement of your work. When I read your intoduction and looked at fig.1, I decided that you consider high precised attitude control dedicated to remote sensing problem or tracking some observation area. But your attitude control just cancels external forces and that's it. So, having no torques acting on the leader satellite (i.e. J(dω/dt) + ω×Jω = 0, when u = d) and considering dynamically symmetrical satellite (from Table 1: J_l(1,1)=J_l(2,2)=18.9), means that the angular motion of leader satellite is regular precession. And so will be with follower satellite. What is the goal of this kind of angular motion?

Can you add zoom in subplots to figures 3, 4 and 6? Now I can't see to what kind of values converges relative q, w and R. I attached the example of what I am asking for.

Author Response

Dear editor and reviwer, our revised reply is provided in the attachment. 

Round 3

Reviewer 2 Report

All my suggestions were met and included to the article.